# OpenReview forum: "Adversarial Robustness of LLM-Based Multi-Agent Systems for Engineering Problems"
_ICLR.cc/2026/Conference — ICLR 2026 Conference Withdrawn Submission_

### Official Review · Reviewer_jLyp · 2025-10-16

**Soundness:** 2
**Presentation:** 2
**Contribution:** 3
**Rating:** 2
**Confidence:** 4

**Summary:**

This paper investigates the adversarial robustness of LLM-based Multi-Agent Systems in engineering contexts. The authors selected several representative tasks, such as pipe pressure loss calculation and mathematical modeling, to explore this topic. Through a series of controlled experiments simulating red-team attacks, they conclude that the vulnerabilities of MAS in engineering domains differ from those in generic contexts and that the design of prompts is critically important for system robustness.

**Strengths:**

- The paper proposes a novel and important problem: the adversarial robustness of MAS in real-world engineering applications. I believe this is indeed a critical issue that warrants investigation.

- The findings on the impact of different prompts, names, and authority levels on the rejection rate are compelling. They provide a valuable reference for the design of MAS specifically for the engineering domain.

- The conclusions of this paper have strong practical implications and contribute significantly to the optimization of MAS for engineering tasks.

**Weaknesses:**

- While the problem the paper aims to solve is urgent, the motivation is not clearly articulated. It remains unclear what the significant differences are between the adversarial robustness of MAS for general purposes and for engineering applications. The authors claim that being misled in engineering workflows is more severe than in other tasks, but they do not experimentally demonstrate this severity; they only show the system's susceptibility to being misled. It is crucial to elaborate on the potential consequences if an engineering MAS is misled. Can these consequences be quantified?

- Figure 1 is unclear and fails to clearly illustrate the entire workflow of the MAS. Furthermore, it contains excessive white space. The authors should optimize this figure to make it more aesthetically pleasing and easier to understand.

- The presentation of the experimental conclusions is disorganized. I suggest that after analyzing each specific task, the authors systematically organize and present the misleading issues encountered by MAS in engineering environments. This could be effectively presented in the main body through a statistical table or detailed case studies.

- This paper reads more like an empirical study, where the authors conduct experiments and present results without engaging in deep discussion or attempting to explain them. The authors need to evaluate more dimensions of the results and provide deeper insights; otherwise, I doubt this work meets the acceptance criteria for a top-tier conference.

- The experiments predominantly use a single model, gpt-4o-mini, with only limited tests on gpt-4o and o3-mini in the appendix. It is unclear whether the conclusions drawn throughout the paper reflect a common flaw in current MAS or are merely limitations specific to the model used.

- The paper is missing some relevant citations, such as G-Safeguard[1] and ARGUS[2], which also discuss the dangers and monitoring of misinformation propagation in MAS.

[1] arxiv.org/abs/2502.11127

[2] arxiv.org/abs/2506.00509

**Questions:**

- I admit that I do not have extensive expertise in traditional engineering. But I would like to know if the four representative problems introduced by the authors can genuinely cover the majority of engineering domains.

- I am curious about the impact of certain hyperparameters on the experimental results, such as the number of interaction rounds, the number of advisors in the MAS, and different topological structures of the MAS. When these parameters change, do the conclusions presented in the paper still hold? I believe this is a very important point regarding the generalizability of the findings.

---

> ### Author Response · Authors · 2025-11-14
>
> Dear Reviewer,
>
> Thank you very much for your constructive feedback. We appreciate the time and care you invested in evaluating our submission.
>
> To briefly address your questions:
>
> Of course, there are almost infinitely many different fields of engineering and our four problems just scratch at a handfull of them. While we cannot include all of them we should have included at least some more of them. But more important than the problems themselves are probably the different ways of introducing subtle errors in the answers and analyzing them consistently.
>
> We agree that especially the different architectures of MAS will probably have major impact on their robustness. From the other hyperparameters, the maximum number of iterations might have impact on some results, where currently there is a high percentage of no decision.
>
> We are grateful for your detailed suggestions regarding motivation, presentation, citations, and experiments, and we will incorporate them as we further develop the project. Although we are withdrawing the current submission, your feedback is very valuable.

---

### Official Review · Reviewer_E51W · 2025-10-22

**Soundness:** 2
**Presentation:** 2
**Contribution:** 2
**Rating:** 2
**Confidence:** 4

**Summary:**

This paper studies how adversarial agents affect LLM-based multi-agent systems in engineering tasks. Through representative problems, the authors show that such systems are especially vulnerable to adversarial influence, depending on task type and agent communication. They also identify design strategies to improve robustness and emphasize the importance of domain-specific evaluation for safe MAS deployment in engineering.

**Strengths:**

1. This paper is easy to follow and is organized well.
2. The findings of this paper may be useful for solving engineering problems using multi-agent systems.
3. This paper has studied the robustness of multi-agent systems under various problems.

**Weaknesses:**

1. The author’s research setting is solving engineering problems, so I believe the choice of model should lean towards more powerful models such as GPT-5 or Gemini-2.5-pro. However, in the main text, the author uses GPT-4o-mini. Furthermore, as shown in Figure 7 of the appendix, both GPT-4o and o3-mini exhibit significantly greater robustness than GPT-4o-mini. This greatly undermines the validity and credibility of the conclusions presented in the main text.

2. The author investigates the robustness of multi-agent systems in the context of solving engineering problems and assumes the existence of malicious agents, but an important aspect is missing: in this setting, how are malicious agents generated? In other words, from a practical perspective, how does this threat model hold up?
    - What are the attackers' intentions?
    - How do attackers manage to poison an agent? Does this require assuming the multi-agent system accepts information from the external environment?
    - What are the concrete harms that the multi-agent system might suffer after an attack?
    - Compared to the threat model in language tasks, what are the fundamental differences in this setting?

    The lack of thorough discussion and analysis in these areas makes it difficult for me to grasp the substantive contribution of this paper relative to prior work.

**Questions:**

See weakness.

---

> ### Author Response · Authors · 2025-11-14
>
> Dear Reviewer,
>
> Thank you very much for your constructive review and for the time you spent evaluating our work.
>
> To briefly address the questions raised:
>
> We agree that using stronger model families together with more complex problems is important for establishing more credible conclusions; However, we believe that the capabilities of the applied model and the complexity of the problem together have a major influence on the outcomings. Thus, stronger models will always find attacks easier in simple problems because it is easier for them to solve the problem. Our simple problems just don't really fit to the stronger LLMs.
>
> We also acknowledge that our threat model requires clearer justification. We believe that in a future where solutions of LLMs in engineering problems will not be validated every time, their adversarial robustness is very important, eg. when designing bridges or dimensioning other safety critical parts/buildings. For the attacks there are different options, one example being online interaction with web sites or other agents for information retrieval.
>
> We appreciate your feedback, even though we are withdrawing the current submission.

---

### Official Review · Reviewer_NDu7 · 2025-11-02

**Soundness:** 2
**Presentation:** 2
**Contribution:** 1
**Rating:** 2
**Confidence:** 4

**Summary:**

This paper presents the first systematic study of adversarial robustness in large-language-model (LLM) based multi-agent systems (MAS) applied to engineering problems. The authors focus on four representative tasks—pipe pressure loss (Darcy-Weisbach), cantilever beam deflection, basic math problems, and Eulerian graph traversal—and inject controlled semantic or numerical errors via a “misleading advisor” agent. This paper evaluates how variations in leader and advisor system prompts, LLM choice and decoding parameters, task formulation and complexity, number, order, and naming of agents affect the MAS’s tendency to accept or reject adversarial suggestions and final accuracy. The findings are: (1) Explicit warnings and non-concise leader prompts dramatically improve rejection rates; (2) Tasks with subtle numeric confusions are most vulnerable; (3) the agent who speaks first exerts outsized influence; (4) Naming or framing advisors as “experts” amplifies their persuasive power..

**Strengths:**

1. This paper is the first to study adversarial prompt-level attacks on LLM-MAS in formal, numerically rigorous domains.

2. All the details are provided in the appendix.

**Weaknesses:**

1. Experiments are confined to idealized, textbook problems (Darcy–Weisbach formula, cantilever beam, simple sums and small graphs). There is no evidence that these findings carry over to complex, real-world engineering workflows (e.g. multiphysics simulations, CAD integration, control-system design).

2. The paper limits itself to a single style of adversarial error (e.g. replacing 64/Re with 25/Re). Modern adversarial-example techniques (gradient-based prompt generation, reinforcement-learning attackers) are omitted. The threat model is therefore narrow and unlikely to capture the true worst-case behaviors of a malicious agent.

3. The authors test ~20 prompt variants without explaining how these were selected or whether they cover the most impactful dimensions of system-prompt design. There is no principled methodology (e.g. pilot-study clustering, mutual-information ranking) to justify prompt choices, raising concerns of cherry-picking.

4. The MAS model is a fixed, synchronous, two-phase, turn-based “leader + advisor(s)” scheme with a hard 5-round limit. Real MAS employ asynchronous messaging, hierarchical committees, or dynamic role reassignment. It is unclear whether the first-mover and “expert” effects persist in more realistic protocols.

5.

**Questions:**

The study falls short in practical relevance, methodological rigor, and threat-model completeness. Before reconsideration, the authors should:

1. Demonstrate transfer of findings to complex, real-world engineering workflows.

2. Incorporate automated adversarial-prompt generation to cover a broader threat space.

3. Apply principled selection of prompt perturbations and control for multiple statistical tests.

4. Explore more realistic MAS protocols (asynchronous, hierarchical).

---

> ### Author Response · Authors · 2025-11-14
>
> Dear Reviewer,
>
> Thank you very much for your review and suggestions for improvement. We appreciate the time and care you put into evaluating our submission.
>
> We agree that our paper is lacking some strong connections to the real world and applies the problem to only too few toy problems , that our MAS mostly consists of of a really simple 2-Agent discussion and that there is room for improvement in the statistic evaluation.
>
> Although we are withdrawing the paper, your feedback will help our future work.
>
> Thank you again for your valuable comments.

---

### Official Review · Reviewer_7Scj · 2025-11-04

**Soundness:** 2
**Presentation:** 2
**Contribution:** 1
**Rating:** 0
**Confidence:** 4

**Summary:**

This paper studies the  adversarial robustness of LLM-based multi-agent systems (MAS) specifically within engineering contexts. The MAS turns out to be a conversation between an LLM solving the problem and another LLM provides adversarial misleading information. Authors argue that  different from generic tasks, robustness in engineering problems is highly sensitive to the task's structural complexity, the subtlety of injected numerical errors, and the communication order. The study also identifies actionable design choices, including prompt framing, agent role assignment, and discussion order, that can significantly improve the trustworthiness of these systems in engineering applications.

**Strengths:**

1. Originality: the studied topic, adversarial robustness in engineering problems seems to be a new domain, and the studied problems are not tested in prior work

2. The paper is easy to understand.

**Weaknesses:**

1. This paper investigates adversarial robustness in **engineering problems**. However, I have concerns about the "engineering problem" setting This paper only studies four problems, namely, pipe pressure loss (Darcy-Weisbach equations), beam deflection, fundamental mathematical modeling, and graph theory. Based on my understanding , the latter two problems should be categorized as mathematical and programming problems. The beam deflection problem has also been studied in previous benchmarks MMLU and GPQA.

How do you define a problem as one of the **engineering problems**, instead of math, code or other problems studied in prior work? And why the adversarial robustness of **engineering problems** has different properties as other MAS? Secondly, only four case studies are sufficient to draw the conclusions.

2. In the method section, it seems the MAS is just a conversation between an LLM solving the problem and another LLM provides adversarial misleading information. Since the second agent does not aim to solve this problem, I think the studied setting is far from really world MAS settings.  In Section 4.3, more advisors setting is discussed. However, it is not clear how these advisors are interacting with other agents.

3. Only OpenAI family LLMs are studied in this work. It is not clear whether the conclusion in this work can transfer to other LLMs, i.e., the robustness issue is GPT4-mini's weakness or a shared weakness of all LLMs.

**Questions:**

1. In Section 3.2, what is Re?

2. In Section 4.1, how are the percentages computed. Run the same system with the same initial prompt multiple times and average the final results?

3. What is the base performance of the 4 tasks without adversary?

---

> ### Author Response · Authors · 2025-11-14
>
> Dear reviewer,
>
> Thank you very much for your constructive review. We appreciate the time you invested in evaluating our work.
>
> To briefly address your questions:
>
> Re refers to the Reynolds number.
>
> The percentages in Section 4.1 are computed by running the same setup multiple times and counting successes and failures divided by the total number of runs.
>
> The base performance of all four tasks without an adversary was not tested by us.
>
> We agree that this paper does not meet the level of the ICLR and are withdrawing the paper.
>
> Thank you for your helpful comments.

---

### Note · Authors · 2025-11-14

I have read and agree with the venue's withdrawal policy on behalf of myself and my co-authors.